# Financial Insecurity and Food Insecurity among U.S. Children with Secondhand and Thirdhand Smoke Exposure

**DOI:** 10.3390/ijerph19159480

**Published:** 2022-08-02

**Authors:** E. Melinda Mahabee-Gittens, Keith A. King, Rebecca A. Vidourek, Ashley L. Merianos

**Affiliations:** 1Division of Emergency Medicine, Cincinnati Children’s Hospital Medical Center, University of Cincinnati College of Medicine, Cincinnati, OH 45229, USA; 2School of Human Services, University of Cincinnati, Cincinnati, OH 45221, USA; keith.king@uc.edu (K.A.K.); rebecca.vidourek@uc.edu (R.A.V.); ashley.merianos@uc.edu (A.L.M.)

**Keywords:** children, tobacco smoke exposure, secondhand smoke, thirdhand smoke, financial insecurity, food insecurity

## Abstract

Objectives: Smokers with financial and food insecurity may find it difficult to quit smoking and reduce their children’s tobacco smoke exposure (TSE). The objective was to examine the associations between child TSE and financial and food insecurity among U.S. school-aged children. Methods: We examined the 2018–2019 National Survey of Children’s Health data on 17,484 children 6–11 years old. Children were categorized into TSE groups: (1) No TSE: did not live with a smoker; (2) thirdhand smoke (THS) exposure alone: lived with a smoker who did not smoke inside the home; or (3) secondhand smoke (SHS) and THS exposure: lived with a smoker who smoked inside the home. We conducted weighted logistic, ordinal, and linear regression analyses to assess the relationships between child TSE status and financial and food insecurity, adjusting for covariates. Results: Overall, 13.1% and 1.8% of children had THS exposure alone and SHS and THS exposure, respectively. Compared to children with no TSE, children with THS exposure alone were at 2.17 increased odds (95% CI = 1.83, 2.58, *p* < 0.001) and children with SHS and THS exposure were at 2.24 increased odds (95% CI = 1.57, 3.19, *p* < 0.001) of having financial insecurity. Children with THS exposure alone were at 1.92 increased odds (95% CI = 1.58, 2.33, *p* < 0.001) and children with SHS and THS exposure were at 2.14 increased odds (95% CI = 1.45, 3.16, *p* < 0.001) of having food insecurity. Conclusions: Children with TSE are at increased risk of experiencing financial and food insecurity. When developing tobacco interventions, a holistic approach to tobacco control that addresses ways to decrease financial and food hardships may improve outcomes.

## 1. Introduction

Despite recent declines in cigarette smoking rates [1], the prevalence of smoking remains disproportionately higher at 27%, in adults who live in poverty, compared to 14% in adults who have higher incomes [1]. Rates of tobacco smoke exposure (TSE) in children from low-income households reflect these same disparities, as children living in impoverished homes have TSE rates as high as 54%, compared to TSE rates of 23% in children living in higher-income homes [2]. Reasons for poverty-driven disparities in adult tobacco use and child TSE among individuals who live in poverty compared to their more affluent counterparts include higher levels of unemployment, stress, nicotine addiction, and decreased access to healthcare, insurance coverage, and tobacco cessation resources [3,4].

Unfortunately, adult smoking cessation and child TSE reduction interventions that have targeted low-income families often yield disappointing results [5,6]. This may be because many tobacco interventions focus on changing tobacco use patterns without addressing difficult issues that families who are living in poverty have to navigate, such as financial insecurity and food insecurity [7,8]. Low-income smokers may experience stress and anxiety as a result of these insecurities, making tobacco abstinence less of a priority in their lives [3,9,10]. Reasons for financial insecurity may include high unemployment rates and high spending on cigarettes [11,12]. In turn, food insecurity may result if smokers preferentially spend money on tobacco products instead of healthy food and essential items, or if they lack income or the means to obtain food [11,12]. If tobacco cessation interventions were to take into account some of the complex life issues that low-income smokers encounter, they may be more successful [9]. For example, if tobacco cessation interventions included ways to help families of smokers to have access to free or reduced-cost foods, this could result in decreased parental stress and anxiety, which could potentially improve tobacco cessation outcomes [13].

When designing smoking cessation and child TSE reduction interventions for low-income families, it is important to address issues related to poverty, financial insecurity, and food insecurity [3,10]. Our study objective was to examine the associations between child home TSE status and financial insecurity, food insecurity, and the number and types of government assistance programs received for food or cash assistance among U.S. children aged 6–11 years. We considered children exposed to tobacco smoke if they lived with a smoker who did not smoke indoors and were exposed to home thirdhand smoke (THS) or aged SHS alone [14], or if they lived with a smoker who smoked indoors and were exposed to both SHS and THS. 

## 2. Materials and Methods

### 2.1. Participants and Procedures

The National Survey of Children’s Health (NSCH) is a cross-sectional survey that collects data on the physical and emotional health and well-being of 0–17-year-old U.S. children. The annual NSCH is conducted by the U.S. Census Bureau in conjunction with the U.S. Health Resources and Services Administration’s Maternal and Child Health Bureau [15,16]. We conducted a secondary analysis of the 2018–2019 NSCH data, including 17,484 children aged 6–11 years; detailed study procedures are described elsewhere [17,18,19]. This study was limited to 6–11-year-old children in order to: (1) exclude adolescents who may have been primary tobacco product users, since NSCH does not include questions to assess if children are active tobacco product users; and (2) examine children who were likely to attend school as one of the measures (i.e., received free/reduced-cost school breakfasts or lunches) did not apply to younger, non-school aged children. A university-based institutional review board deemed the present study as being “not human subjects” research and exempt from review.

### 2.2. Measures

#### 2.2.1. Child Home TSE Status

Parents were asked if their child lived with any household members that smoked tobacco (e.g., cigarettes). If parents responded “yes” to this question, then they were asked if anyone smokes inside their home. These two questions were combined to create the child’s home TSE status, which was categorized into the three levels: (1) no TSE: child did not live with a smoker; (2) THS exposure alone: child lived with a smoker who did not smoke inside the home. This criterion was based on research indicating that children can still be exposed to THS even when they live in homes with smoking bans and they are not around active smokers [20,21]; and (3) SHS and THS exposure: child lived with a smoker who smoked inside the home. This criterion was based on research that children who live with smokers who actively smoke indoors can be exposed to both SHS and THS [14].

#### 2.2.2. Financial Insecurity

Parents were asked: “Since this child was born, how often has it been very hard to cover the basics, such as food and housing on your family’s income [22,23]”? Response options were never (0), rarely (1), somewhat often (2), or very often (3) hard to get by on family income. Higher scores were indicative of increased financial insecurity; response option 0 served as the reference category in this study.

#### 2.2.3. Food Insecurity

Parents were asked: “Which of these statements best describes your household’s ability to afford the food you need during the past 12 months [22,23]”? Response options were “we could always afford to eat good nutritious meals” (0), “we could always afford enough to eat but not always the kinds of food we should eat” (1), “sometimes we could not afford enough to eat” (2), and “often we could not afford enough to eat” (3). Higher scores were indicative of increased food insecurity; response option 0 served as the reference category in this study.

#### 2.2.4. Number and Types of Government Assistance Programs Received for Food or Cash Assistance

Parents were asked: “At any time during the past 12 months, even for one month, did anyone in your family receive”: (1) “Cash assistance from a government welfare program”; (2) “Food Stamps or Supplemental Nutrition Assistance Program (SNAP) benefits”; (3) “Free or reduced-cost breakfasts or lunches at school”; and (4) “Benefits from the Woman, Infants, and Children (WIC) Program”. All four yes/no items were assessed separately and compositely to assess the number of programs received (range 0–4) [24].

#### 2.2.5. Covariates

The following sociodemographic characteristics were included as covariates due to associations with TSE in prior research [2,25]: child age; sex; race/ethnicity (non-Hispanic White, non-Hispanic Black, Hispanic, non-Hispanic Other/Multiracial); parent education level (≤high school graduate and equivalent, some college, ≥college degree); family household structure (two currently married parents, two not currently married parents, single parent, other family type); and family federal poverty level (FPL; 0–199%, 200–299%, 300–399%, and ≥400%). NSCH provided a calculated variable for federal poverty level for public use based on the State Children’s Health Insurance Program (SCHIP) income groups [24].

### 2.3. Statistical Analysis

The 2018–2019 NSCH methodology guidelines were followed, which included applying sampling weights to account for NSCH survey nonresponses, possible sampling frame issues, and to match survey responses with the U.S. child population [24]. Descriptive statistics including weighted percentages for all variables of interest were performed. Weighted chi-square tests were performed to examine the relationships between the covariates and child home TSE status, with the exception of child age where a one-way analysis of variance (ANOVA) test was performed. The covariates, except child age and child sex, were significantly associated with child home TSE status. Therefore, we conducted a series of weighted logistic, ordinal, and linear regression analyses, depending on the nature of the dependent variable, to assess the relationships between child home TSE status and financial insecurity and food insecurity, while adjusting for the covariates. For assessing the relationship between child TSE status and the number and types of government assistance programs received for food or cash assistance, all covariates but FPL were included in the linear and logistic regression models since this variable is used to determine eligibility for government programming. We excluded all missing cases prior to analyses and a two-sided *p*-value, with *p* < 0.05 indicating significance, was used. Analyses were conducted using SPSS Complex Samples version 28.0 [24].

## 3. Results

The mean (standard error, SE) age of the 17,484 children aged 6–11 years was 8.56 (0.03) years, and 51.0% were male; 50.8% were non-Hispanic White, 13.3% were non-Hispanic Black, 24.7% were of Hispanic origin, and 11.2% were non-Hispanic Other race including Multiracial (Table 1). A total of 13.1% of children were exposed to THS alone and 1.8% were exposed to both SHS and THS. 

### 3.1. Child and Family Covariates and Child Home TSE Status

There were significant differences between child race/ethnicity, parent education level, family household structure, and FPL and child home TSE status (Table 1). Regarding child race/ethnicity and home TSE status, we observed the highest rates of THS exposure alone among non-Hispanic White children (58.2%) followed by Hispanic children (22.2%), non-Hispanic Other or Multiracial children (10.1%), and non-Hispanic Black children (9.5%). The highest rates of SHS and THS exposure were among children who were non-Hispanic White (62.5%), followed by non-Hispanic Black children (23.0%), non-Hispanic Other or Multiracial children (8.4%), and Hispanic children (6.1%). Children with THS exposure alone and children with SHS and THS exposure had higher percentages of parents who had a lower education of ≤high school graduate or equivalent (38.4% and 57.6%, respectively) compared to children with no TSE (24.8%). Concerning family household structure and home TSE status, 53.7% of children with THS exposure alone and 25.2% of children with SHS and THS exposure lived in homes with two currently married parents compared to 67.6% of children who had no TSE. Children with THS exposure alone and children with SHS and THS exposure had higher percentages of being in the lower FPL of 0–199% (53.7% and 82.0%, respectively) compared to children with no TSE (27.3%).

### 3.2. Child Home TSE Status and Financial and Food Insecurity

Concerning financial insecurity and child TSE, children with THS exposure alone were at 2.17 increased odds (95% CI = 1.83, 2.58, *p* < 0.001) and children with SHS and THS exposure were at 2.24 increased odds (95% CI = 1.57, 3.19, *p* < 0.001) of having financial insecurity compared to children with no TSE, while adjusting for all covariates (Table 2). 

Concerning food insecurity, children with THS exposure alone were at 1.92 increased odds (95% CI = 1.58, 2.33, *p* < 0.001) and children with SHS and THS exposure were 2.14 increased odds (95% CI = 1.45, 3.16, *p* < 0.001) of having food insecurity compared to children with no TSE, while adjusting for all covariates (Table 2).

### 3.3. Child Home TSE Status and Number and Types of Food or Cash Assistance Items Received from the Government

Child home TSE status was significantly associated with the number of food or cash assistance items received in the past 12 months. Children with THS exposure alone (β = 0.27, 95% CI = 0.16, 0.38, *p* < 0.001) and children with SHS and THS exposure (β = 0.60, 95% CI = 0.42, 0.77, *p* < 0.001) received a higher number of food or cash assistance items compared to children with no TSE, while adjusting for the covariates (Table 3). The types of cash and food assistance are found in Table 4.

A total of 6.0% of children with THS exposure alone and 7.8% of children with SHS and THS exposure received cash assistance from the government. There was no significant difference between child home TSE status and receipt of cash assistance from the government.

A total of 30.7% of children with THS exposure alone and 59.2% of children with SHS and THS exposure received food stamps or SNAP benefits. Children with THS exposure alone were at 2.25 increased odds (95% CI = 1.75, 2.90, *p* < 0.001) and children with SHS and THS exposure were at 4.83 increased odds (95% CI = 3.12, 7.46, *p* < 0.001) of receiving food stamps or SNAP benefits compared to children with no TSE, while adjusting for the covariates.

A total of 52.8% of children with THS exposure alone and 79.3% of children with SHS and THS exposure received free/reduced-cost school breakfasts or lunches. Children with THS exposure alone were at 1.90 increased odds (95% CI = 1.53, 2.36, *p* < 0.001) and children with SHS and THS exposure were at 3.95 increased odds (95% CI = 2.49, 6.27, *p* < 0.001) of receiving free/reduced-cost school breakfasts or lunches compared to children with no TSE, while adjusting for child age, child sex, child race/ethnicity, parent education level, and family household structure.

A total of 11.6% of children with THS exposure alone and 14.6% of children with SHS and THS exposure received WIC program benefits. There was no statistically significant difference between child home TSE status and receiving WIC program benefits.

## 4. Discussion

The results of this study indicate that 6–11-year-old children who live in homes in which they are exposed to THS alone, or exposed to both SHS and THS, are at increased odds of having financial insecurity, food insecurity, and receiving different forms of government assistance including food stamps or SNAP benefits and free/reduced-cost school breakfasts or lunches. Notably, children with the highest rates of either THS alone or SHS and THS exposure were non-Hispanic White and non-Hispanic Black, had parents with lower education levels, and lived in homes with a single parent and with lower FPLs. These higher TSE rates based on race/ethnicity, education levels, and income are congruent with those found in prior research [25,26].

We observed high poverty rates in children with TSE. While NSCH does not provide public access to income levels, we report that 54% of children with THS exposure alone, and 82% of children with SHS and THS exposure, had the lowest (0–199%) FPL. Childhood poverty is associated with numerous adverse mental and physical health outcomes [27]. The chronic stress and myriad issues associated with poverty may play a role in children’s brain development as children living in impoverished homes may have adversely affected neurocognitive development in the areas of executive functioning, language, and cognition [27,28]. Poverty is also associated with adverse health effects including increased inflammatory markers, and increased future risks of cardiovascular disease and obesity [29]. In parallel to the morbidity associated with childhood poverty, much research indicates that child TSE is also associated with similar adverse acute and long-term health effects [30,31]. Thus, it is difficult to disentangle whether TSE or poverty individually are the main drivers of these observed outcomes or if the combination of TSE and poverty and the associated factors (e.g., crowding, stress) [27] are responsible for these adverse outcomes. Nevertheless, there is a clear need to reduce TSE in children who live in poverty in order to decrease short- and long-term TSE-related consequences.

We found high levels of financial and food insecurity in children with TSE, which is not surprising given the high levels of poverty in the families of children with TSE [2,25,26]. Concerning receipt of government food assistance, a positive finding is that children with TSE were at increased odds of receiving food stamps and free or reduced-cost school breakfast or lunches, which indicates that these families are receiving government assistance that will potentially relieve some of their food insecurity and the associated stressors [32]. While children with TSE were not at increased odds to receive cash assistance, this is likely because only about 3% of all children, irrespective of their home TSE status, received this type of assistance. However, research suggests that some families may not apply for assistance due to the associated stigma or fear or mistrust of child welfare services [33]. Lack of observed differences between child home TSE status and WIC program benefits may be because we examined 6–11-year-old children who were not eligible for WIC past the age of five years [34].

These results suggest that assisting parents who smoke to secure government resources is needed to potentially decrease their financial and food insecurities. While it is encouraging that children with TSE were at increased odds to receive food stamps and free or reduced cost school breakfasts or lunches, it is also important to be cognizant that families who are just above the U.S. poverty-based threshold for receiving government assistance may still be in need of aid of resources such as subsidized housing and home energy assistance [35]. Such assistance may then improve smokers’ access to better health insurance and cessation resources and could potentially facilitate their engagement in tobacco cessation programs [3,36]. Thus, when developing interventions and policies to decrease the prevalence of adult smoking and child TSE, the American Academy of Pediatrics recommends framing tobacco use and dependence using a structural competency “lens” [4]. This is because of the myriad of forces, such as poverty, financial insecurity, and food insecurity, that interplay in smokers’ lives to make it harder to quit tobacco use. The forces affecting low-income smokers include specific tobacco marketing strategies [4]; higher rates of unemployment, which results in stress, financial and food insecurity [3]; lack of or poor health insurance and access to care [3]; and lack of enforcement of smoke-free housing rules [37]. Thus, it is likely that if factors such as these are not considered when developing future tobacco cessation interventions, the disparities in tobacco use and TSE will continue to be perpetuated in certain adult and child subgroups, and especially in those of lower socioeconomic status.

This study has strengths including the use of two waves of NSCH data that is nationally representative of 6–11-year-old children and the use of many measures to assess financial hardships. However, some limitations should be acknowledged. The NSCH is a cross-sectional survey and, thus, causal associations cannot be ascertained, nor can any longitudinal trends or outcomes be assumed or determined. Child TSE was assessed by parent report and SHS or THS was not biochemically verified; TSE verification would have been optimal to confirm the validity of parent reports. Parents also reported their income levels, financial and food insecurity, and receipt of government assistance. Although these measures allowed us to assess several aspects of financial hardship, parent reports could have been subject to reporting or recall biases and some of the responses may have referred to other household members. Nevertheless, these parent reports may have also led to underreporting, which may have resulted in even stronger observed associations.

## 5. Conclusions

In conclusion, we found high rates of poverty, financial insecurity, and food insecurity in families with children with home TSE, compared to children without TSE. The use of government assistance in families with home TSE were higher than those without home TSE, but there is room for improvement. Given these findings, when developing tobacco cessation interventions, it is important for researchers to be cognizant that a more holistic approach to tobacco control that goes beyond focusing on nicotine addiction in low-income families is needed [3]. Thus, prior to developing and testing tobacco control interventions and policies to protect and reduce pediatric TSE, professionals should keep population groups that have the highest rates of TSE in mind and consider including intervention components that address smokers’ nicotine addiction and the contextual factors in their lives that may make quitting tobacco use seem like an impossible task. These components may include ways to help with increasing their access to healthy food and stress management strategies, as these tactics may ultimately improve tobacco abstinence and TSE reduction outcomes that tobacco treatment programs are targeting [10].

## Figures and Tables

**Table 1 ijerph-19-09480-t001:** Sociodemographic Characteristics and Child Home TSE Status among U.S. Children 6–11 Years Old, 2018–2019 NSCH.

Characteristic		Child Home TSE Status			*p*-Value ^b^
	Overall (N = 17,484)	No Home TSE (*n* = 14,882)	THS Exposure Alone (*n* = 2295)	SHS and THS Exposure (*n* = 307)	
	*n* (%) ^a^	*n* (%) ^a^	*n* (%) ^a^	*n* (%) ^a^	
**Child Age, *M* (SE)**	8.56 (0.03)	8.55 (0.03)	8.60 (0.07)	8.68 (0.18)	0.670
**Child Sex**					0.603
Male	9097 (51.0)	7756 (51.3)	1171 (49.2)	170 (53.1)	
Female	8387 (49.0)	7126 (48.7)	1124 (50.8)	137 (46.9)	
**Child Race/Ethnicity**					**<0.001**
Non-Hispanic White	11,985 (50.8)	10,116 (49.5)	1657 (58.2)	212 (62.5)	
Non-Hispanic Black	1129 (13.3)	975 (13.6)	106 (9.5)	48 (23.0)	
Hispanic	2129 (24.7)	1865 (25.5)	250 (22.2)	14 (6.1)	
Non-Hispanic Other or Multiracial	2241 (11.2)	1926 (11.5)	282 (10.1)	33 (8.4)	
**Parent Education Level**					**<0.001**
≤High school graduate/equivalent	2643 (27.1)	1908 (24.8)	593 (38.4)	142 (57.6)	
Some College	4254 (22.6)	3235 (20.7)	891 (33.1)	128 (33.6)	
≥College Degree	10,587 (50.3)	9739 (54.5)	811 (28.5)	37 (8.8)	
**Family Household Structure**					**<0.001**
Two parents, currently married	12,104 (65.0)	10,777 (67.6)	1227 (53.7)	100 (25.2)	
Two parents, not currently married	1223 (8.3)	875 (7.8)	297 (11.3)	51 (12.8)	
Single parent	3342 (21.3)	2640 (20.0)	593 (27.7)	109 (35.7)	
Other family type	815 (5.4)	590 (4.6)	178 (7.3)	47 (26.4)	
**Family Federal Poverty Level**					**<0.001**
0–199%	5024 (40.2)	3831 (27.3)	982 (53.7)	211 (82.0)	
200–299%	2922 (16.1)	2400 (15.9)	467 (18.0)	55 (11.6)	
300–399%	2536 (12.3)	2240 (12.9)	278 (9.6)	18 (3.3)	
≥400%	7002 (31.4)	6411 (34.0)	568 (18.7)	23 (3.1)	

Abbreviations: NSCH, National Survey on Children’s Health; TSE, tobacco smoke exposure; THS, thirdhand smoke exposure; SHS, secondhand smoke exposure; M, mean; SE, standard error. ^a^
*n* refers to raw counts and percentages are weighted column percentages, unless noted otherwise. ^b^ Bold font indicates statistical significance *p* < 0.05.

**Table 2 ijerph-19-09480-t002:** Child Home TSE Status and Financial Insecurity and Food Insecurity among U.S. Children aged 6–11 Years, 2018–2019 NSCH.

	Financial Insecurity	Ordinal Regression			Food Insecurity	Ordinal Regression		
	*M* (SE) ^a^	AOR	95% CI	*p*-Value ^b^	*M* (SE) ^a^	AOR	95% CI	*p*-Value ^c^
**Home TSE Status**								
No TSE	0.64 (0.02)	Ref	Ref	Ref	0.38 (0.02)	Ref	Ref	Ref
THS Exposure Alone	0.96 (0.04)	**2.17**	**1.83, 2.58**	**<0.001**	0.56 (0.03)	**1.92**	**1.58, 2.33**	**<0.001**
SHS and THS Exposure	1.00 (0.08)	**2.24**	**1.57, 3.19**	**<0.001**	0.64 (0.07)	**2.14**	**1.45, 3.16**	**<0.001**
**Child Age, *M* (SE)**	-	1.00	0.96, 1.04	0.869	-	0.99	0.94, 1.03	0.605
**Child Sex**								
Male	0.86 (0.04)	Ref	Ref	Ref	0.51 (0.03)	Ref	Ref	Ref
Female	0.88 (0.04)	1.05	0.92, 1.19	0.513	0.55 (0.03)	1.16	0.99, 1.35	0.064
**Child Race/Ethnicity**								
Non-Hispanic White	0.89 (0.03)	Ref	Ref	Ref	0.52 (0.03)	Ref	Ref	Ref
Non-Hispanic Black	0.83 (0.05)	0.86	0.69, 1.06	0.153	0.54 (0.04)	1.09	0.87, 1.37	0.465
Hispanic	0.90 (0.04)	1.01	0.82, 1.23	0.948	0.51 (0.04)	1.01	0.81, 1.25	0.944
Non-Hispanic Other or Multiracial	0.86 (0.04)	0.87	0.71, 1.05	0.146	0.54 (0.04)	1.07	0.84, 1.36	0.590
**Parent Education Level**								
≤High school graduate/equivalent	0.83 (0.04)	1.14	0.94, 1.39	0.182	0.54 (0.04)	**1.50**	**1.21, 1.86**	**<0.001**
Some College	0.97 (0.04)	**1.56**	**1.33, 1.84**	**<0.001**	0.59 (0.03)	**1.83**	**1.54, 2.16**	**<0.001**
≥College Degree	0.80 (0.04)	Ref	Ref	Ref	0.45 (0.03)	Ref	Ref	Ref
**Family Household Structure**								
Two parents, currently married	0.79 (0.04)	Ref	Ref	Ref	0.48 (0.03)	Ref	Ref	Ref
Two parents, not currently married	0.93 (0.06)	**1.41**	**1.06, 1.87**	**0.018**	0.56 (0.04)	**1.35**	**1.02, 1.80**	**0.038**
Single parent	1.03 (0.04)	**1.77**	**1.48, 2.11**	**<0.001**	0.60 (0.03)	**1.54**	**1.28, 1.86**	**<0.001**
Other family type	0.72 (0.05)	0.84	0.65, 1.09	0.185	0.47 (0.05)	0.99	0.74, 1.32	0.954
**Family Federal Poverty Level**								
0–199%	1.11 (0.04)	**4.00**	**3.35, 4.77**	**<0.001**	0.73 (0.03)	**5.98**	**4.68, 7.64**	<0.001
200–299%	0.98 (0.04)	**3.20**	**2.68, 3.83**	**<0.001**	0.60 (0.03)	**4.50**	**3.51, 5.78**	<0.001
300–399%	0.80 (0.05)	**2.08**	**1.72, 2.52**	**<0.001**	0.45 (0.03)	**2.47**	**1.89, 3.22**	<0.001
≥400%	0.58 (0.04)	Ref	Ref	Ref	0.34 (0.03)	Ref	Ref	Ref

Abbreviations: NSCH, National Survey on Children’s Health; TSE, tobacco smoke exposure; THS, thirdhand smoke exposure; SHS, secondhand smoke exposure; M, mean; SE, standard error; AOR, adjusted odds ratio; CI, confidence interval; Ref, reference category. ^a^
*M* (SE) refers to mean (SE) financial insecurity and mean (SE) food insecurity with higher scores indicative of higher financial insecurity (range 0 (“never hard to get by on family income”) to 3 (“very often hard to get by on family income”)) and higher food insecurity (range 0 (“we could always afford to eat good nutritious meals”) to 3 (“often we could not afford enough to eat”)), respectively. ^b^ Ordinal regression model with reference category as “never hard to get by on family income” and adjusting for the covariates of child age, child sex, child race/ethnicity, parent education level, family household structure, and federal poverty level. Bold font indicates statistical significance *p* < 0.05. ^c^ Ordinal regression model with reference category as “we could always afford to eat good nutritious meals” and adjusting for the covariates of child age, child sex, child race/ethnicity, parent education level, family household structure, and federal poverty level. Bold font indicates statistical significance *p* < 0.05.

**Table 3 ijerph-19-09480-t003:** Child Home TSE Status and Number of Instances of Cash or Food Assistance Items Received among U.S. Children aged 6–11 Years, 2018–2019 NSCH.

	Number of Cash/Food Assistance Received (Range 0–4)	Multiple Linear Regression		
	*n* (%) ^a^	Beta ^b^	95% CI	*p* Value
**Home TSE Status**				
No TSE	0.93 (0.03)	Ref	Ref	Ref
THS Exposure Alone	1.20 (0.05)	**0.27**	**0.16**, **0.38**	**<0.001**
SHS and THS Exposure	1.53 (0.09)	**0.60**	**0.42**, **0.77**	**<0.001**
**Child Age, *M* (SE)**	-	**−0.02**	**−0.04**, **−0.01**	**0.001**
**Child Sex**				
Male	1.22 (0.04)	Ref	Ref	Ref
Female	1.22 (0.04)	−0.01	−0.05, 0.05	0.975
**Child Race/Ethnicity**				
Non-Hispanic White	0.99 (0.04)	Ref	Ref	Ref
Non-Hispanic Black	1.53 (0.05)	**0.54**	**0.46**, **0.63**	**<0.001**
Hispanic	1.28 (0.06)	**0.29**	**0.20**, **0.38**	**<0.001**
Non-Hispanic Other or Multiracial	1.08 (0.04)	**0.09**	**0.03**, **0.15**	**0.006**
**Parent Education Level**				
≤High school graduate/equivalent	1.5 (0.04)	**0.66**	**0.42**, **0.55**	**<0.001**
Some College	1.3 (0.05)	**0.49**	**0.58**, **0.74**	**<0.001**
≥College Degree	0.83 (0.04)	Ref	Ref	Ref
**Family Household Structure**				
Two parents, currently married	0.93 (0.04)	Ref	Ref	Ref
Two parents, not currently married	1.28 (0.06)	−0.06	−0.19, 0.08	0.394
Single parent	1.30 (0.04)	−0.08	−0.025, 0.10	0.385
Other family type	1.36 (0.07)	**−0.43**	**−0.56**, **−0.30**	**<0.001**

Abbreviations: NSCH, National Survey on Children’s Health; TSE, tobacco smoke exposure; THS, thirdhand smoke exposure; SHS, secondhand smoke exposure; M, mean; SE, standard error; CI, confidence interval; Ref, reference category. ^a^ *M* (SE) refers to mean (SE) number of detracting elements (range 0–3). ^b^ Multiple linear regression model adjusting for the covariates of child age, child sex, child race/ethnicity, parent education level, and family household structure. Bold font indicates statistical significance *p* < 0.05.

**Table 4 ijerph-19-09480-t004:** Child Home TSE Status and Types of Cash and Food Assistance Received among U.S. Children aged 6–11 Years, 2018–2019 NSCH.

	Received Cash Assistance	Multivariable Logistic Regression			Received Food Stampsor SNAP Benefits	Multivariable Logistic Regression			Received Free/Reduced-Cost School Breakfasts or Lunches	Multivariable Logistic Regression			Received WIC Program Benefits	Multivariable Logistic Regression		
	*n* (%) ^a^	AOR	95% CI	*p*-Value ^b^	*n* (%) ^a^	AOR	95% CI	*p*-Value ^b^	*n* (%) ^a^	AOR	95% CI	*p*-Value ^b^	*n* (%) ^a^	AOR	95% CI	*p*-Value ^b^
**Home TSE Status**																
No TSE	292 (2.8)	Ref	Ref	Ref	1282 (14.4)	Ref	Ref	Ref	3341 (32.7)	Ref	Ref	Ref	555 (7.7)	Ref	Ref	Ref
THS Exposure Alone	95 (6.0)	1.81	0.97, 3.37	0.060	526 (30.7)	**2.25**	**1.75**, **2.90**	**<0.001**	1010 (52.8)	**1.90**	**1.53**, **2.36**	**<0.001**	136 (11.6)	1.41	0.96, 2.07	0.08
SHS and THS Exposure	28 (7.8)	1.31	0.65, 2.66	0.448	148 (59.2)	**4.83**	**3.12**, **7.46**	**<0.001**	221 (79.3)	**3.95**	**2.49**, **6.27**	**<0.001**	30 (14.6)	1.67	0.82, 3.39	0.157
**Child Age**, ***M* (SE)**	8.50 (0.15)	0.96	0.86, 1.06	0.401	8.46 (0.07)	**0.91**	**0.86**, **0.96**	**0.002**	8.61 (0.05)	1.00	0.95, 1.05	0.998	8.13 (0.11)	**0.83**	**0.76**, **0.90**	**<0.001**
**Child Sex**																
Male	207 (3.1)	Ref	Ref	Ref	1041 (17.6)	Ref	Ref	Ref	2.457 (37.3)	Ref	Ref	Ref	386 (8.2)	Ref	Ref	Ref
Female	208 (3.6)	1.22	0.84, 1.76	0.303	915 (17.0)	1.05	0.86, 1.29	0.630	2115 (34.9)	0.93	0.79, 1.09	0.372	335 (8.4)	1.08	0.82, 1.42	0.605
**Child Race/Ethnicity**																
Non-Hispanic White	191 (2.1)	Ref	Ref	Ref	937 (11.0)	Ref	Ref	Ref	2.412 (23.5)	Ref	Ref	Ref	320 (4.3)	Ref	Ref	Ref
Non-Hispanic Black	82 (7.3)	**2.39**	**1.47**, **3.89**	**<0.001**	381 (39.4)	**3.77**	**2.95**, **4.83**	**<0.001**	626 I61.6)	**3.95**	**3.15**, **4.97**	**<0.001**	118 (13.3)	**2.86**	**2.01**, **4.08**	**<0.001**
Hispanic	87 (4.1)	1.70	0.90, 3.20	0.100	383 (20.6)	**1.50**	**1.12**, **2.02**	**0.007**	963 (52.5)	**2.70**	**2.17**, **3.36**	**<0.001**	182 (14.4)	**2.72**	**1.92**, **3.85**	**<0.001**
Non-Hispanic Other or Multiracial	55 (2.8)	1.26	0.73, 2.18	0.399	255 (13.0)	1.26	0.95, 1.66	0.106	571 (27.5)	**1.29**	**1.04**, **1.60**	**0.020**	101 (7.0)	**1.75**	**1.23**, **2.49**	**0.002**
**Parent Education Level**																
≤High school graduate/equivalent	150 (6.0)	**2.44**	**1.57**, **3.79**	**<0.001**	804 (34.0)	**6.03**	**4.66**, **7.81**	**<0.001**	1539 (63.7)	**5.90**	**4.78**, **7.28**	**<0.001**	255 (15.3)	**4.29**	**2.99**, **6.17**	**<0.001**
Some College	148 (4.7)	**2.22**	**1.38**, **3.58**	**0.001**	781 (24.9)	**4.29**	**3.34**, **5.51**	**<0.001**	1740 (50.7)	**4.35**	**3.65**, **5.17**	**<0.001**	266 (11.6)	**3.66**	**2.62**, **5.12**	**<0.001**
≥College Degree	117 (1.3)	Ref	Ref	Ref	371 (4.9)	Ref	Ref	Ref	1293 (14.8)	Ref	Ref	Ref	200 (3.0)	Ref	Ref	Ref
**Family Household Structure**																
Two parents, currently married	98 (1.5)	Ref	Ref	Ref	553 (8.1)	Ref	Ref	Ref	1956 (24.0)	Ref	Ref	Ref	360 (6.6)	Ref	Ref	Ref
Two parents, not currently married	47 (4.9)	**2.32**	**1.13**, **4.80**	**0.002**	266 (30.3)	**2.95**	**2.05**, **4.26**	**<0.001**	521 (55.6)	**2.26**	**1.65**, **3.11**	**<0.001**	100 (14.6)	1.45	0.92, 2.31	0.113
Single parent	163 (5.2)	**2.29**	**1.355**, **3.91**	**0.022**	930 (36.8)	**3.69**	**2.91**, **4.68**	**<0.001**	1604 (57.7)	**2.46**	**2.01**, **3.01**	**<0.001**	209 (10.3)	0.93	0.66, 1.30	0.658
Other family type	107 (15.8)	**7.45**	**4.24**, **13.08**	**<0.001**	207 (32.0)	**2.16**	**1.55**, **3.00**	**<0.001**	491 (67.8)	**3.11**	**2.31**, **4.19**	**<0.001**	52 (11.0)	0.91	0.56, 1.50	0.714

Abbreviations: Abbreviations: NSCH, National Survey on Children’s Health; SNAP, Supplemental Nutrition Assistance Program; WIC, Woman, Infants, and Children; TSE, tobacco smoke exposure; THS, thirdhand smoke exposure; SHS, secondhand smoke exposure; M, mean; SE, standard error; AOR, adjusted odds ratio; CI, confidence interval; Ref, reference category. ^a^ *n* refers to unweighted sample size and % refers to weighted row percentage, unless otherwise noted. ^b^ Multivariable logistic regression models with reference category as “no” and adjusting for the covariates of child age, child sex, child race/ethnicity, parent education level, and family household structure. Bold font indicates statistical significance *p* < 0.05.

## Data Availability

Data are publicly available on the NSCH website at https://www.childhealthdata.org/learn-about-the-nsch/NSCH (accessed on 24 January 2022).

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
