# Peer review of "Financial Insecurity and Food Insecurity among U.S. Children with Secondhand and Thirdhand Smoke Exposure"

_ijerph, 2022, doi:10.3390/ijerph19159480_

Round 1

Reviewer 1 Report

 I think the results provided are really interesting and should be published. The manuscript is well structured, bybliography it is really updated  and relevant.

I just have a small suggestion that could be interesting to include. The National survey  of children’s Health is a cross-sectional survey that collects data from 0 to 17 years old in the US. Maybe,It would be really interesting to provide information available in the other age groups (apart from the group from 6 to 11 years). It could be especially relevant because todlers are really more exposed to THS and teens could have been actively smokers.

Reviewer 2 Report

This manuscript examines the association between child tobacco smoke exposure (TSE) and food insecurity and financial insecurity, important issues facing low-income families. Secondary analyses were conducting using cross-sectional data from a large, national survey including two cohorts.  The authors adjusted multivariable models for covariates previously shown to be associated with child TSE, including federal poverty level, race and ethnicity, and child’s age and sex.  Further, the author’s examined the associations among TSE and number/type of government assistance received.  Overall, this paper will well-written and provides a compelling case for understanding financial and food insecurity in relation to TSE.

Introduction

The introduction provides a clear overview of the relevant literature and a rationale for the proposed study.

The second paragraph of the introduction makes the case for the association between financial insecurity and tobacco use. However, I have questions regarding the logical flow of these ideas. For example, is it accurate to say that “smokers may experience financial insecurity due to …. and high spending on cigarettes.”  The cited studies do indicate that smokers may prioritize cigarette spending over other necessities, but is there a probable causal link?  Then following this argument that cessation may not be a priority, would it directly follow that providing support for purchasing food could reduce parental stress and anxiety, which then increases cessation rates?  Perhaps the authors could address the linkages between stress, anxiety, financial and food insecurity, and cessation earlier in this paragraph?

Please brief state the rationale for focusing on the 6 -11 year old age group.

Material and Methods

Line 80, I believe the authors’ meant that the IRB deemed the present study “exempt” from review rather than “example.”

Could the authors address the potential method of third hand smoke exposure – presumably from clothing if the smoker in the home does not take steps to reduce the child’s exposure? Alternately, the authors could a indicate a relevant reference for this category (or the categories in general).

Results

The paragraph “3.1. Child and Family Covariates and Child Home TSE Status” reports the findings in Table 1 by column. However, the second and third sentences as currently written seems to imply that the rates of THS exposure and SHS and THS exposure were 58% and 62% for non-Hispanic White, and that the percentages were rates of exposure for Hispanic and non-Hispanic Black children.  From my reading, these should be reported as column percentages.

Discussion

The discussion clearly summarizes the findings and includes relevant literature to provide context to the results.

The third paragraph uses the language that the children received assistance, however, the methods state that these questions referred to anyone in the household receiving any of the types of assistance.  This clarification through the paragraph is particularly important given that none of the children including in these analyses are eligible for WIC benefits due to their age.

The authors include in their discussion that “these results suggest assisting parents who smoke to secure government resources may decrease their financial and food insecurities.” However, the analyses do not appear to address this question.  Families with children exposed to TSE had higher financial and food insecurities and also were more likely to receive assistance, however, none of the analyses indicate lower insecurities among those receiving assistance.  If the authors are able to provide analyses showing this association, this conclusion would be appropriate. However, questions remain about whether the government assistance is enough to offset financial and food insecurities or even whether some of the families would meet eligibility criteria (e.g., those who are just above the threshold for assistance but struggle to provide food and meet their expenses).
